# Hybrid longitudinal-transverse phonon polaritons

Christopher R. Gubbin [1], Rodrigo Berte [2,3], Michael A. Meeker[4], Alexander J. Giles[4], Chase T. Ellis[4], Joseph G. Tischler[4], Virginia D. Wheeler[4], Stefan A. Maier[3], Joshua D. Caldwell [5] & Simone De Liberato [1]

Phonon polaritons, hybrid light-matter quasiparticles resulting from strong coupling of the electromagnetic field with the lattice vibrations of polar crystals are a promising platform for mid-infrared photonics but for the moment there has been no proposal allowing for their electrical pumping. Electrical currents in fact mainly generate longitudinal optical phonons, while only transverse ones participate in the creation of phonon polaritons. We demonstrate how to exploit long-cell polytypes of silicon carbide to achieve strong coupling between transverse phonon polaritons and zone-folded longitudinal optical phonons. We develop a microscopic theory predicting the existence of the resulting hybrid longitudinal-transverse excitations. We then provide an experimental observation by tuning the resonance of a nanopillar array through the folded longitudinal optical mode, obtaining a clear spectral anti-crossing. The hybridisation of phonon polaritons with longitudinal phonons could represent an important step toward the development of phonon polariton-based electrically pumped mid-infrared emitters.

---

[1] School of Physics and Astronomy, University of Southampton, Southampton SO17 1BJ, UK. [2] CAPES Foundation, Ministry of Education of Brazil, Brasilia, DF 70040-020, Brazil. [3] Chair in Hybrid Nanosystems, Nanoinstitute Munich, Faculty of Physics, Ludwig-Maximilians-Universität München, 80539 München, Germany. [4] U.S. Naval Research Laboratory, Washington, DC 20375, USA. [5] Department of Mechanical Engineering, Vanderbilt University, Nashville, Tennessee 37205, USA. Correspondence and requests for materials should be addressed to S.D.L. (email: s.de-liberato@soton.ac.uk)

Phonon polaritons are mixed light-matter excitations arising from hybridisation between photons and transverse optical phonons in polar dielectrics. In the crystal's Reststrahlen band, between the transverse optical (TO) and longitudinal optical (LO) phonon frequencies, the real part of the dielectric function is negative. In these spectral window these resonances result in electric fields which are strongly localised at the crystal surface, leading to the appearance of localised modes termed surface phonon polaritons (SPhPs).

Following initial studies on surfaces[1,2] or in waveguides[3], SPhPs in user defined nanostructures were demonstrated[4,5]. Such localised fields allow for energy confinement on length-scales orders of magnitude shorter than that of the free photon wavelength[6], without the strong optical losses associated with plasmonic systems[7,8]. These modes are also extremely tunable, thanks to their morphologic nature[9], their dependence on carrier density[10,11] and their ability to hybridise with propagating[12] or epsilon-near-zero modes[13]. Recent investigations have demonstrated the potential of localised phonon polaritons for sensing[14], nonlinear optics[15–18], waveguiding[19], nanophotonic circuitry[20] and rewritable nano-optics[21,22].

The tunable, narrowband nature of SPhP resonances makes them a good candidate for realisation of integrated mid-infrared emitters. To this end phonon polariton-based thermal emitters have been demonstrated[2,23,24], but thermal pumping is intrinsically inefficient and does not allow for an increase in the degree of temporal coherence. In polaritonic systems based on electronic excitations, electrical injection of polaritonic modes has been demonstrated[25,26], but similar schemes with phonon polaritons are difficult to implement as their energies typically lie an order of magnitude below the electrical bandgap. Electrical currents do however couple efficiently to crystal lattice vibrations. In fact one of the main sources of Ohmic loss in polar dielectrics is through LO phonon emission via the Fröhlich interaction[27]. The use of such an interaction to pump SPhPs is however problematic as the LO phonon frequency defines the upper edge of the Reststrahlen band, which limits the spectral overlap and resonant transitions between the LO phonon and SPhPs. More fundamentally the photonic field, due to its transverse nature, does not couple with longitudinal excitations. We are thus at an impasse: we can efficiently create large populations of LO phonons via electrical currents, but only TO phonons can form SPhPs and emit light in the far-field.

In this paper we theoretically predict and experimentally demonstrate an approach to strongly couple LO phonons with SPhPs, exploiting silicon carbide (SiC) polytypes whose unit cells are elongated along the c-axis. This extension of the atomic lattice inserts an additional Bragg plane in the direction of the c-axis, folding the phonon dispersion back to the Γ point. These zone-folded LO phonons (ZFLOs) can then become resonant with the SPhPs at optical wavelengths, as illustrated in Fig. 1a. The mechanical boundary conditions for the nuclear displacement at the crystal interface can mix the two excitations, leading to novel hybrid modes arising from the strong coupling of SPhPs and ZFLO phonons. The resulting quasiparticles, which we name Longitudinal-Transverse Phonon Polaritons (LTPP) possess both a longitudinal character, which could potentially allow resonant generation by Ohmic losses[27], and a transverse one, making far-field emission possible[23].

The ZFLO modes we exploit, typically termed weak phonons, are high-wavevector states accessible near the Γ point due to Bragg scattering induced by the periodicity of the crystal lattice. They manifest as a dip in planar reflectance and are usually phenomenologically described by adding oscillators to the material's transverse dielectric function[28,29]. The negative dispersion of the LO phonon ensures that these weak phonon modes exist within the Reststrahlen band, co-existing in frequency with propagating or localised SPhPs[17]. This is illustrated in Fig. 1a for 4H-SiC, the material studied in this Letter, whose weak phonon lies at around 837.5/cm. Around 250 unique polytypes of SiC exist, each with different weak phonon frequencies, allowing the weak LO phonon to be tuned throughout the Reststrahlen region. The weak phonons of 15R- and 6H-SiC for example lie near 860 and 885/cm, respectively.

Longitudinal-transverse hybridisation has been also theoretically predicted in polar quantum wells and superlattices[30,31], and realised in plasmonic systems, where nanoscale confinement of transverse plasmonic modes makes very large wavevectors accessible, intersecting the negative dispersion of the longitudinal oscillation of the electron gas[32,33]. This results in red shift of the modal frequency, which can become non-negligible when an appreciable portion of the plasmonic field exists at large wavevectors. Contrastingly in the systems under investigation Bragg folding ensures that the longitudinal mode is accessible for all values of the wavevector, which as we will

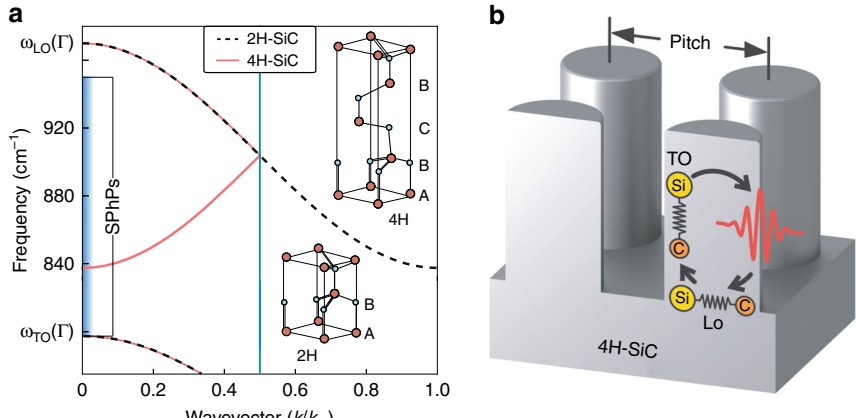

**Fig. 1** Schematic of the ZLFO-SPhP hybridisation scheme. **a** Illustration of LO phonon dispersion parallel to the c-axis in 2H- and 4H-SiC[29]. The wavevector is normalised over the 2H-SiC Brillouin zone border $k_M = \pi/a$ where $a$ is the length of the 2H-SiC unit cell along the c-axis. The shaded region illustrates the spectral range where SPhPs exist at a SiC/vacuum interface. Inset shows an illustration of the crystal structures of 2H- and 4H-SiC, the length of the 4H- unit cell is approximately twice that of the 2H- along the c-axis. **b** Sketch of the strong coupling between LO phonons, TO phonons, and photons, resulting in the creation of LTPPs, in a square array of 4H-SiC nanopillars

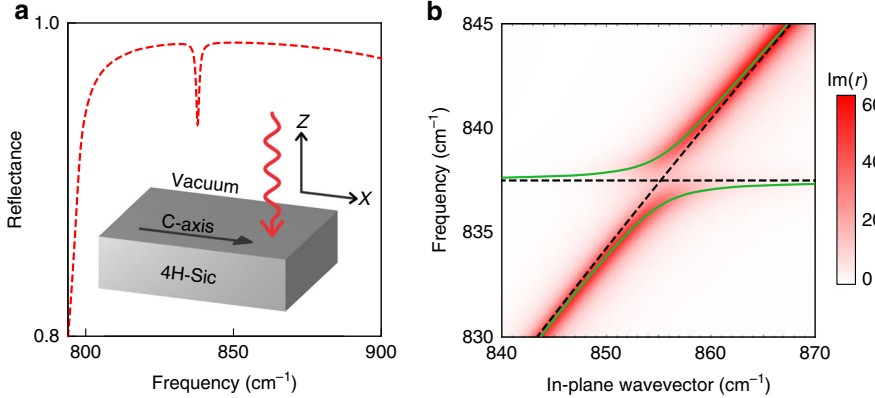

**Fig. 2** Coupling between a ZFLO and a SPhP resonance at the interface of a polar dielectric halfspace. **a** TM polarised reflectance calculated for an a-cut, 4H-SiC substrate supporting a weak phonon mode at near normal incidence. The in-plane wavevector of the incident light is parallel to the crystal c-axis. The inset illustrates the physical system under study. **b** Colormap illustrates the dispersion of the surface phonon polariton on the bilayer interface calculated by Eq. (4). Overlaid dashed lines indicate the bare weak phonon and surface phonon polariton dispersions, while the green curves illustrate the coupled LTPP modes

show leads to a strong hybridisation even in optically large resonators.

In the following we will initially develop a microscopic theory of light-matter coupling in polar dielectric systems including spatial dispersion. This theory will be then used to theoretically investigate the reflectance of a 4H-SiC surface. Finally, we will present experimental results demonstrating strong coupling, and thus the existence of LTPP, using arrays of 4H-SiC nanopillars.

## Results

**Theory**. Our starting point in order to microscopically model the hybridisation of phonon polaritons with ZFLOs is to expand the theory describing ionic motion in a polar dielectric[34–36] to the retarded regime. In frequency-space the material displacement **X** obeys the equation

$$\left[\omega_T^2 - \omega(\omega + i\gamma)\right]\mathbf{X} = -\beta_L^2 \nabla(\nabla \cdot \mathbf{X}) + \beta_T^2 \nabla \times \nabla \times \mathbf{X} \\ - \frac{\alpha}{\rho}(\nabla\phi - i\omega\mathbf{A}),$$ (1)

where $\phi(\mathbf{A})$, is the electromagnetic scalar (vector) potential, the material high-frequency dielectric constant is $\varepsilon_\infty$, the transverse (longitudinal) optical phonon frequency at the $\Gamma$ point is $\omega_T(\omega_L)$, the material density is given by $\rho$, the phonon damping rate by $\gamma$, the transverse (longitudinal) phonon velocities in the limit of quadratic dispersion by $\beta_T(\beta_L)$ and the polarizability $\alpha$. In this we assume that the only effect of the anisotropy is the Bragg folding along the c-axis.

In section 1 of the Supplementary Note we solve Eq. (1), in conjunction with the Maxwell equations by the introduction of auxiliary scalar and vector potentials $Y = \nabla \cdot \mathbf{X}$, $\mathbf{\Theta} = \nabla \times \mathbf{X}$, allowing us to write the ionic displacement as a hybrid, containing both transverse and longitudinal components whose mixing will be instigated by application of the appropriate mechanical and Maxwell boundary conditions

$$\mathbf{X} = \frac{1}{\omega_T^2 - \omega(\omega+i\gamma)}\left[\beta_T^2 \nabla \times \mathbf{\Theta} - \frac{\beta_L^2 \varepsilon_\infty}{\varepsilon(\omega,0)}\nabla Y \\ - \frac{\alpha}{\rho}(\nabla\phi_H - i\omega\mathbf{A})\right],$$ (2)

in which $\phi_H$ is the homogeneous electric scalar potential, solution to the Laplace equation, and $\varepsilon(\omega, 0)$ is the lattice dielectric function in the absence of spatial dispersion. Including spatial dispersion it

is easy to show that the full dielectric function is given by

$$\varepsilon(\omega, \mathbf{k}) = \varepsilon_\infty \frac{\omega_L^2 - \omega(\omega + i\gamma) - \beta_T^2 k^2}{\omega_T^2 - \omega(\omega + i\gamma) - \beta_T^2 k^2},$$ (3)

for the transverse fields.

**Application to SPhPs**. In order to clearly demonstrate how our theory leads to the appearance of hybrid LTPP modes, here we apply it to the analytically solvable case of an a-cut uniaxial polar dielectric halfspace in vacuum, shown in the inset of Fig. 2a. We choose an a-cut crystal because in this system the ZFLO phonon manifests as a dip in the planar reflectance, permitting us to compare our analytical solution with experimental observation of ZFLOs previously reported in the literature[4,28,29,37]. In section 2 of the Supplementary Note we apply the formalism outlined in the previous section to this geometry, applying the appropriate electrical and mechanical boundary conditions and calculate the Fresnel coefficient for TM polarised light incident along the c-axis, including spatial dispersion, to be

$$r = \frac{k_{zB} - \frac{k_{zT}}{\varepsilon(\omega, \mathbf{k}_T)} - \Omega}{k_{zB} + \frac{k_{zT}}{\varepsilon(\omega, \mathbf{k}_T)} + \Omega},$$ (4)

where $k_{zB}$ ($k_{zT}$) are the out-of-plane wavevector components of the transverse mode in the vacuum (dielectric) defined as

$$k_{zB}^2 = \frac{\omega^2}{c^2} - k_x^2,$$ (5)

$$k_{zT}^2 = \frac{\omega_{TO}^2 - \omega^2}{v_T^2} - k_x^2,$$ (6)

the vector $\mathbf{k}_T = (k_x, 0, k_{zT})$, $c$ is the speed of light in vacuum, $v_T$ is a characteristic velocity describing the TO phonon dispersion and $\Omega$ encodes the mechanical boundary condition $\bar{\sigma} \cdot \mathbf{z}|_{z=0} = 0$, where $\bar{\sigma}$ is the stress tensor. It is important to note that Eq. (4) cannot be reduced to a simple equation where the dispersionless dielectric function $\varepsilon(\omega)$ is replaced by the spatially dispersive $\varepsilon(\omega, \mathbf{k})$ derivable from Eq. (1). This is because additionally to the effect of spatial dispersion calculating the reflectance now involves application of both mechanical and Maxwell boundary conditions, resulting in a mixing dependent on the wavevectors on each side of the boundary. We can apply this result to the description of the ZFLO modes observed in planar reflectance measurements of a-cut SiC polytypes by shifting the in-plane wavevector of the

longitudinal component $k_{xL}$ inside $\Omega$

$$k_{xL} = \frac{2\pi}{a} + k_x, \tag{7}$$

while leaving the wavevectors of the transverse components unchanged. Here $a$ is the length of the unit cell along the $c$-axis and $k_x$ is the in-plane wavevector of the incident photons. The result for a 4H-SiC substrate is shown in Fig. 2a. The anisotropy of such a polytype can be taken into account by noting that the leading terms in the numerator and denominator of Eq. (4) when neglecting the ZFLO mode ($\Omega = 0$) are just those from the Fresnel coefficient of an isotropic halfspace. We can then replace the term for the lower halfspace with that derived for an uniaxial halfspace in the local approximation by considering the transverse wavevector to be that of the extraordinary wave in the crystal, yielding a characteristic dip in the reflectance at the ZFLO frequency 837.5/cm, consistent with previously reported experimental data[29,37].

This result also allows for investigation of the guided modes of the planar structure, satisfying

$$k_{zB} + \frac{k_{zT}}{\varepsilon(\omega, \mathbf{k}_T)} + \Omega = 0, \tag{8}$$

whose dispersion is seen in Fig. 2b, where the imaginary component of the reflectance coefficient Eq. (4) is plotted utilising standard parameters for the 4H-SiC dielectric function with damping rate $\gamma = 4$/cm. The clearly visible spectral anticrossing between the dispersion of SPhPs supported on the planar interface and the comparatively dispersionless ZFLO phonon is the hallmark of strong coupling. It demonstrates that, close to resonance, the bare modes (black dashed lines in Fig. 2b) hybridise, creating two novel spectrally resolved hybrid longitudinal-transverse quasiparticle branches which we named LTPP (green solid lines).

**Experimental results.** In order to verify the existence of LTPP, we consider square arrays of cylindrical 4H-SiC resonators on a same-material c-cut substrate[4,12,38]. Such systems, sketched in Fig. 1b, support a variety of transverse SPhP modes, with highly tuneable frequencies dependent on the geometrical parameters[9,12]. The monopole mode in particular, polarised out-of the substrate plane, is highly sensitive to the interpillar spacing (pitch) due to the repulsion of like charges on adjacent pillars and can effectively be tuned throughout the Reststrahlen band as has been shown in previous studies[9,38]. This mode is often referred to as longitudinal in the literature but this naming convention only refers to the electric field orientation with respect to the pillar long axis, the mode is nonetheless electromagnetically transverse, with non-vanishing curl. Apart from its technological relevance thanks to small mode volumes and narrow and tuneable resonances, this system presents a key advantage over the planar system discussed in the previous section. The SPhPs here exist within the light-line, and it is thus possible to spectroscopically probe the anticrossing without the need for complex prism coupling set-ups[39].

The underlying mechanism that gives rise to the longitudinal-transverse hybrid mode in the theoretical treatment of the polar dielectric halfspace and in the micropillar arrays we experimentally probed is the same. As such we expect to observe a spectral anticrossing in the micropillar array resonances, essentially analogous to the one shown in Fig. 2b, but entirely contained within the light-line. However, the nanostructured array complicates the theoretical analysis since there is no analytical solution to Eq. (1) for this system. The experimental practicality of the nanopillar array comes in fact at the cost of heavy numerical complications, which dramatically increase the computational power required to solve the problem. In order to tackle the problem numerically the electromagnetic fields, described by Maxwell's equations, must be coupled through the mechanical boundary conditions to the ionic equation of motion in Eq. (1). In contrast to the the planar case considered in the previous section where solutions were Bloch modes for which the in-plane wavevector was a good quantum number, the micropillar modes contain many wavevectors whose values will be affected by their coupling to the longitudinal degrees of freedom as in plasmonic nonlocality[33]. Given the importance of mechanical boundary conditions for the longitudinal-transverse hybridisation, prohibitively expensive full 3D simulations of the coupled electromagnetic and material displacement fields are thus necessary to describe LTPP in the resonator array on substrate system.

We exploit the broad tunability of the monopole mode by fabricating samples with interpillar spacings in the range 700–2000 nm, over which the resonance is expected to tune from the high to the low energy side of the ZFLO phonon at 837.5/cm. Pillars were fabricated with a uniform height of 950 nm and diameters of 300 and 500 nm. Nanopillar arrays were fabricated from semi-insulating c-cut 4H-SiC substrates by reactive ion etching[4]. We probe the planar reflectance utilising Fourier transform infrared spectroscopy.

Results are shown in Fig. 3 for pillars of nominal diameters 300 nm (a) and 500 nm (b) for the full range of interpillar spacings explored. A larger diameter results in a blue shift of the monopolar mode[9], while increasing the interpillar spacing causes the monopolar mode to red shift as a result of decreased coupling between resonators. When the monopolar mode approaches the 4H-SiC ZFLO, illustrated by the horizontal dashed line in Fig. 3, a second branch appears in the reflectance on the low energy side of the ZFLO mode. Rather than continuing to red shift through the ZFLO for large interpillar spacings, the monopolar mode remains on it's high energy side and the new branch red shifts. This anticrossing behaviour, previously illustrated in Fig. 1, is a hallmark of strong coupling and demonstrates that the monopolar phonon polariton mode of the pillar array is hybridised with the ZFLO phonon[12,13].

Further evidence for the hybrid nature of the observed LTPP resonances can be acquired from the magnitude of the recorded reflectance dips, calculated by subtracting the reflectance of the pillar array from that of the planar substrate, as shown in the lower panel of Fig. 3. In this panel the blue (red) circles correspond to the upper (lower) branches in the upper panel. For small or large interpillar spacings, where the detuning between monopolar and ZFLO modes is large we see that, as expected, the upper and lower branches have characteristics of the bare modes. The monopolar mode of the resonator array couples well to the impinging light resulting in a deep reflectance dip, while the ZFLO couples weakly. In the intermediate region instead, the modes are linear combinations of the bare monopolar and ZFLO modes, leading to the crossing for the reflectance dip which demonstrates their hybrid nature. In the lower panel of Fig. 3 the shaded regions indicate where the upper (lower) branch is more ZFLO (monopolar) in character. Particularly the equalisation of the branches reflectance is a signal of the anticrossing point, indicating that both branches are composed of equal parts monopolar and ZFLO modes. This can be seen by following the vertical lines up onto the reflectance maps, corresponding to the avoided crossing of the LTPP branches.

Additional samples were fabricated with the same nominal array parameters. Wide inter-sample variability ensures that these arrays have different monopolar frequencies. All samples reproduce the anticrossing at the ZFLO frequency, this data is available in section 3 of the Supplementary Note.

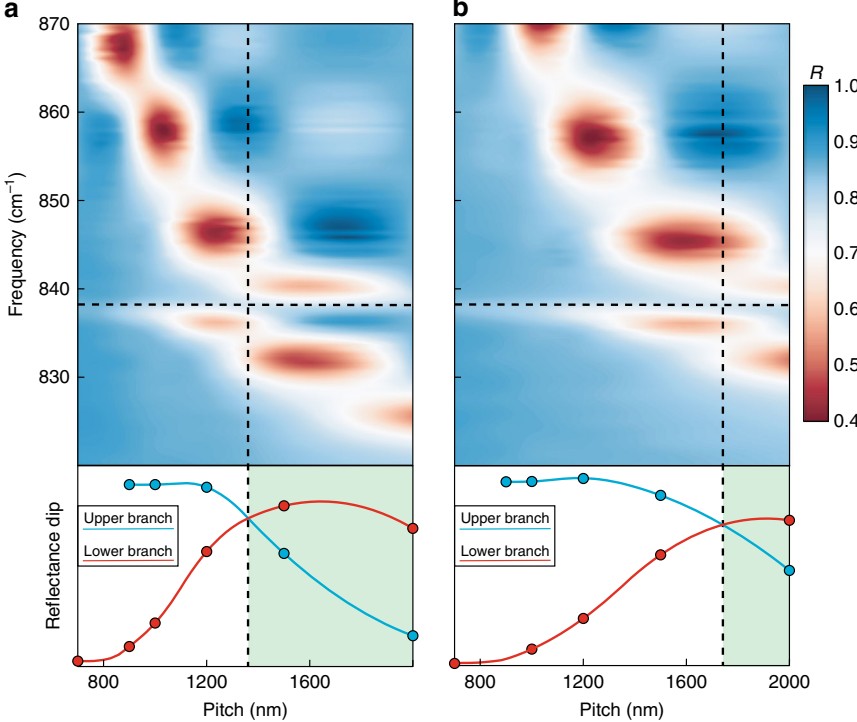

**Fig. 3** Experimental demonstration of strong coupling between a ZFLO and the monopolar mode of a nanopillar array. Upper panels illustrate the experimental reflectance from an array of 4H-SiC nanopillars with diameter (**a**) 300 nm (**b**) 500 nm and height 950 nm recorded as a function of the lattice period. The horizontal dashed line indicates the weak phonon resonance. The lower panel shows the magnitude of the reflectance dip extracted from the data. The shaded region indicates the region where the upper polariton is predominantly LO in character, demarcated by the vertical dashed line

## Discussion

Our results illustrate the hybridisation of longitudinal and transverse modes in polar dielectric structures, thus providing the first clear experimental evidence of the LTPP modes theoretically predicted by our theory. This hybridisation, mediated by the mechanical boundary conditions at the crystal surface, cannot be achieved in bulk as the longitudinal mode cannot be matched without an interface. This kind of surface-induced hybridisation is well understood in plasmonic systems, where spatial dispersion arises as a result of electron pressure[33]. In plasmonic systems however these effects are only accessible where the electric field is confined on the nanoscale, meaning that resonances are comprised of sufficiently high-wavevector Fourier components to experience the dispersion[32]. In the polar dielectric systems discussed here these large wavevector Fourier components are instead accessible in optically large resonators meaning that the hybridisation is essentially accessible in any appropriately tuned polar dielectric resonator.

Fabrication of resonators whose eigenmodes are linear superpositions of transverse and longitudinal waves has important technological implications. Such modes could be directly pumped electrically through the Fröhlich interaction, providing way toward the realisation of phonon polariton-based mid-infrared emitters. Furthermore our simulations show that for hybridised LTPPs non-radiative and radiative losses are of the same order, leading to an appreciable radiative efficiency and potentially allowing for the creation of efficient electroluminescent devices operating throughout the SiC Reststrahlen band. An efficient injection scheme could also potentially lead to the development of coherent phonon polariton-based light sources, an idea which has received some attention in recent literature[40,41]. Further flexibility can be found by applying these results to superlattice systems in which the Brillouin folding can be finely tuned[42] and the hybrid material dielectric function can be controlled[43], potentially

allowing for the creation of electroluminescent devices operating across the mid-infrared spectral region.

## Methods

**Nanofabrication of SiC nanopillar arrays.** Pillar arrays were fabricated by etching into deep semi-insulating 4H-SiC substrates. The pillar geometry was defined using Al/Cr hard masks deposited via electron-beam lithography, lift-off and evaporation. The exposure time of a following reactive ion etch determined the pillar height. The masked substrate was exposed for 38 min at 150 W utilising SF6 and Ar in equal partial pressures, followed by a chemical wet etch. To remove any residual fluorine, a commercial PlasmaSolv treatment was performed.

**Fourier transform infrared spectroscopy.** Infrared measurements were performed in the reflectance mode of a Thermo Scientific, Nicolet FTIR Continuum microscope. A 15×, 0.58 NA reverse Cassegrain objective provided illumination at angles of 10–35° off-normal, with weighted average of 25°. Spectra were taken as an average of 32 scans with 0.5 cm$^{-1}$ resolution acquired from a 50 μm$^2$ area

## Data availability

The data that support the findings of this study are available from the corresponding author upon reasonable request.

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

## Acknowledgements
S.D.L. is a Royal Society Research Fellow. S.D.L and C.R.G. acknowledge support from the Innovation Fund of the EPSRC Programme EP/M009122/1. C.T.E. and J.G.T. acknowledge support from the Office of Naval Research. M.A.M. and C.T.E. acknowledge support from the National Research Council Research Associateship program. R.B. acknowledges the Capes Foundation for a Science Without Borders fellowship (Bolsista da Capes, Proc. No. BEX 13.298/13-5). S.A.M. acknowledges the DFG Cluster of Excellence Nanoinitiative Munich (NIM) and the Bavarian Solar Technologies Go Hybrid (Soltech) programme.

## Author contributions
S.D.L. conceived and led the project. S.D.L. and C.R.G. developed the theory, analysed the data, and wrote the paper. J.G.T., S.A.M., J.D.C. and S.D.L. conceived and designed the experiments. A.J.G., V.D.W. and J.D.C. designed and fabricated the nanostructures. R.B., M.A.M., C.T.E. and J.D.C. all performed FTIR measurements of the nanostructure arrays. All authors discussed the results and commented on the manuscript.
