## [Peer Review File · Nature Communications]

Reviewers' Comments:

Reviewer #1:

Remarks to the Author:

Gubbin et al. reported a scheme of engineering Longitudinal-Transverse Phonon Polariton by hybridizing LO phonon with surface phonon polariton in SiC. They first established the theoretical formalism, which was later corroborated by experiments in both planar geometry and fabricated nanopillar arrays. Anticrossing behavior was observed in the reflection spectra of both samples. This is a nice work exploiting the hybridization of different modes of excitation in mid-infrared spectrum range. The physical picture is clear and the experimental observation is convincing. Especially, tuning such hybridization through engineering the geometric properties of the nanopillar array provides a convenient way of harnessing similar metamaterials for optical applications in mid-infrared, which is a very important yet underdeveloped range. My only concern is Fig. 3. A color scale is needed for the upper panels. This holds true for Fig. 2b as well.

Reviewer #2:

Remarks to the Author:

In their paper, the authors report on a mixed state of hybrid longitudinal-transverse phonon polaritons. This is a very original work where the usual phonon modes (LO and TO) must be redefined and where the LO mode couples radiatively to light. This goal is achieved by patterning SiC into nanopillars and proved by a telltale anticrossing behavior of the resonance of the pillar array with the phonon mode at 838cm^{-1} . The paper should be published in Nature communications, but could be improved by some modifications

- In the abstract, the last sentence seemed to imply there are no mid-infrared electrically pumped emitters - this sentence should be reworded.
- One important implication of the work is the possibility to use LO phonons for radiative emission using electrical driving current. The authors should be more specific in how efficient the process should be. Presumably, the main decay mechanism of the LO phonon will remain anharmonic decay and not radiative emission? This point should be addressed.

Reviewer #3:

Remarks to the Author:

The manuscript reports on the experimental study of novel longitudinal-transversal phonon-polaritons. The authors perform the far-field (mid-IR) spectral measurements of the periodic array of SiC pillars, in which monopolar resonance has anti-crossing with the folded longitudinal optical phonon. The obtained results sound to be of a significant and broad interest and open many interesting possibilities for building optoelectronic devices, where phonons are explored via their electrical pumping.

The paper is very well motivated in the abstract/introduction, and the main idea is more or less clearly stated. However, I was not able to follow the rest of the paper, in which the main results are exposed. Therefore, only after the paper has been improved in lines with the comments I put below, I will be able to judge whether it is suitable for Nature Communications.

1. Sometimes, the authors use unprecise language. For instance, in the introduction I read “these resonances can be strongly localised at the crystal surface”. I guess the authors refer to the localized electromagnetic fields associated with the resonances.

2. In Fig 1 the schematics of the periodic lattice of pillars is shown next to the folded dispersion of LO phonons. Then the Brillouin zone (BZ) is mentioned. It was difficult for me to understand whether the authors referred to the BZ of the atomic lattice or pillar lattice. This should be clarified.

3. It would be really helpful if the authors could add to Fig. 1b the dispersion of both TO phonons and that of phonon-polaritons (for a single SiC-air interface). Otherwise, it is difficult to have a feeling regarding the hybridization of the modes (“strong coupling”) that the authors are aiming to study.

4. I do not understand the significance of Eqs. (1)-(3) for the paper. How do the authors make use of them? What does the displacement X (and equation for it) help me to learn/understand? Initially, I thought that Eq. (3) was used to extract the nonlocal dielectric function $\epsilon(\omega, k)$. In this case, maybe it would prefer to see the explicit expression for $\epsilon(\omega, k)$, rather than the equation for X . Otherwise, for instance, I do not see any connection between Eq. (4) and Eq. (3).

5. I was confused with the notations for the wavevectors. There are 4 of them: k , k_x , k_B and k_T . I guess with k_B and k_T the z -component is meant. But then I would define them as k_{zB} and k_{zT} , explicitly writing up the expressions for them. Does k present the x -component of the wavevector? Then I would simply use k_x . Actually, the notation k_x is used in Eq. (5), so are k and k_x equivalent?

6. In which equation do the authors make the shift of k_x , specified by the Eq. (5)? Do they substitute Eq. (5) into Eq. (4)? This is not mentioned and should be clarified.

7. I do not understand what is meant by the phrase “the anisotropy is reintroduced by considering the transverse wavevector to be that of the extraordinary wave in the crystal”. First of all, is Eq. (4) written for the isotropic half-space? This is not mentioned. Second, I do not understand what the “reintroduction” of the anisotropy means and to which equation/equations it should be applied. Why doing this reintroduction? More details are needed in my opinion.

8. As far as I understand, the authors fail in developing the theory/simulations for their pillar structure because of the complexity of the problem. But is there any chance for the authors to complement their experimental results with some simplified modelling? For example, at least showing the monopole mode of a single rod (in the local approximation). Then the band-structure of the pillar lattice can be roughly represented by the back-banding of the dispersion of the monopole mode. Otherwise, it is really difficult to interpret the results presented in Fig. 3 and be convinced that the author's arguments are valid.

With many of commercially-available (and also free) simulators, based on the finite-elements methods, it does not seem to be very demanding to simulate an infinite periodic array of rods in the local approximation. Actually, the nonlocal approximation can be also roughly taken into account, by plugging in into Eq. 14 (from the supplement) the momentum of the hybridized mode, estimated from the experiment.

9. The authors mention first the "monopole mode", but then talk about interpillar dipole-dipole coupling. It sounds a bit contradicting. Shouldn't it be the monopole-monopole coupling?

10. What are the vertical dashed line in Figs. 3a,b? Do they indicate the position of the monopole resonance. If so, how are the authors can be sure about that without having any reference (for example, from the simulations).

Answer to the Referees

We thank the three Referees for their time and effort, for their positive comments and constructive criticisms. We hope the novel version of the manuscript and of the Supplementary Information will address all their doubts and they will now suggest our work for publication.

REFEREE 1

My only concern is Fig. 3. A color scale is needed for the upper panels. This holds true for Fig. 2b as well.

We thank the Referee for this comment and agree that this would improve the paper's presentation. Color scales have been added for both Fig 2b and Fig 3 as requested.

REFEREE 2

In the abstract, the last sentence seemed to imply there are no mid-infrared electrically pumped emitters - this sentence should be reworded.

We thank the Referee for this comment. We have reworded the abstract to make clear that we are referring to surface phonon polariton based mid-infrared electrically pumped emitters.

Old Text: This represents an important first step in the development of electrically pumped mid-infrared emitters.

New Text: This represents an important first step in the development of surface phonon polariton based electrically pumped mid-infrared emitters.

One important implication of the work is the possibility to use LO phonons for radiative emission using electrical driving current. The authors should be more specific in how efficient the process should be. Presumably, the main decay mechanism of the LO phonon will remain anharmonic decay and not radiative emission? This point should be addressed.

Our paper focuses on the existence of LTPP modes, and we thus decided to defer more detailed discussions on the exploitation of LTPPs for optoelectronic devices to future works. Still we agree with the Referee that an estimate of the radiative efficiency of LTPP modes is important to judge the potential impact of our work. Should such an efficiency be negligible, one of the selling arguments of our results would indeed be moot.

In order to provide such an estimate we start by neglecting ZFLO modes, which allows us to employ standard FEM simulations. We thus use COMSOL to simulate an infinite square lattice of nanopillars with height 900nm, diameter 300nm and interpillar spacing of 600nm, which are characteristic of one of the samples used in the paper. This results in a nanopillar monopolar resonance near to 860/cm.

The finite linewidth Γ_m of the monopolar mode is a result of loss by phonon anharmonic decay in SiC and photon emission to free space, which occur on timescales τ_{ph} and τ_γ respectively. The total linewidth of the monopolar mode is a linear sum of the inverse of these characteristic times

$$\Gamma_m = \frac{1}{\tau_{\text{ph}}} + \frac{1}{\tau_\gamma}. \quad (1)$$

In order to quantify the radiative and non-radiative emission fraction of the monopolar mode we separate the dielectric function of the nanopillars into real and imaginary components as

$$\epsilon(\omega) = \epsilon'(\omega) + ix\epsilon''(\omega), \quad (2)$$

where ϵ' is the real and ϵ'' is the imaginary component of the dielectric function. The variable x is used to artificially change the imaginary component of the dielectric function. In our simulations we tune x over the range $[0.1, 1]$ in order

FIG. 1: Plot of the monopolar linewidth $\Gamma_m(x)$ as a function of the variable x

to extract the characteristic scattering rates for phonon scattering and photon emission. Note that this procedure leads to non-Kramers-Kronig dielectric functions for $x \neq 1$. This procedure is acceptable here, as it is equivalent to estimate the derivative of Γ over x at $x = 1$ and then performing a linear approximation. The total linewidth Γ_m is shown as a function of frequency by the solid line in Fig. 1.

It is clear that the linewidth has a linear dependance on the tuning parameter x . We now fit the data in Fig. 1 with the equation

$$\Gamma_m(x) = \frac{x}{\tau_{\text{ph}}} + \frac{1}{\tau_{\gamma}}. \quad (3)$$

For the geometry considered this yields the results

$$\frac{1}{\tau_{\text{ph}}} = 176.5\text{cm}^{-1}, \quad (4)$$

$$\frac{1}{\tau_{\gamma}} = 194.1\text{cm}^{-1}. \quad (5)$$

Near to the anti-crossing observed in the experiment we contend that the monopolar mode is hybridised to the band-folded LO phonon of the 4H-SiC substrate. At the anti-crossing the polaritonic quasiparticles are composed of roughly equal fractions of monopolar mode and LO phonon, and the polaritonic linewidth Γ_{\pm} is simply given by

$$\Gamma_{\pm} = \frac{1}{2} [\Gamma_{\text{LO}} + \Gamma_m], \quad (6)$$

where Γ_m is the linewidth of the bare monopolar mode considered above and Γ_{LO} is that of the folded LO phonon mode, due to anharmonic decay. At zone centre the SiC LO phonon linewidth is usually found to be about twice that of the TO phonon [1] and under this assumption we can assert that at the anticrossing around 25% of the polariton loss is radiative.

Although we stress this is only an order of magnitude estimate, it shows that an appreciable fraction of the polaritonic loss is through photonic, radiative channels. Even if we decided not to insert the previous discussion into the manuscript, we made the following addition to our conclusions:

New Text: Furthermore our simulations show that for hybridised LTPPs non-radiative and radiative losses are of the same order, leading to an appreciable radiative efficiency and potentially [allowing for the creation of efficient electroluminescent devices operating throughout the SiC Reststrahlen band.]

REFeree 3

1. Sometimes, the authors use unprecise language. For instance, in the introduction I read “these resonances can be strongly localised at the crystal surface”. I guess the authors refer to the localized electromagnetic fields associated with the resonances.

We apologise if the Referee found our language to be imprecise. We have tried to correct this as much as possible throughout. For the specific example which the Referee brought our attention to we made the change detailed below.

Old Text: In this spectral window these resonances can be strongly localised at the crystal surface leading to the appearance of localised modes termed surface phonon polaritons (SPhPs).

New Text: In this spectral window these resonances result in electric fields which are strongly localized at the crystal surface, leading to the appearance of localized modes termed surface phonon polaritons (SPhPs).

2. In Fig 1 the schematics of the periodic lattice of pillars is shown next to the folded dispersion of LO phonons. Then the Brillouin zone (BZ) is mentioned. It was difficult for me to understand whether the authors referred to the BZ of the atomic lattice or pillar lattice. This should be clarified.

We appreciate that this point is confusing and thank the Referee for bringing it to our attention. Whenever a Brillouin zone is referred to in the paper it is that of the crystal lattice. We clarified this in the caption of Fig. 1, stating that the Brillouin zone boundary shown is that of 2H-SiC.

The resonator mode dispersion is not as a result of folding induced by either the lattice or array periodicity. Predominantly dispersive effects arise from the inter pillar coupling (see for example Gubbin et al. Physical Review Letters **116**, 246402 (2016)).

Old Text: In this paper we demonstrate how these problems can be solved, exploiting Bragg zone-folded LO phonons (ZFLO) in silicon carbide (SiC) polytypes whose unit cells are elongated along the c-axis.

New Text: In this paper we demonstrate how these problems can be solved, exploiting silicon carbide (SiC) polytypes whose unit cells are elongated along the c-axis. This extension of the atomic lattice inserts an additional Bragg plane in the direction of the c-axis, folding the phonon dispersion back to the Γ point as illustrated in Fig. 1a. We term these folded modes zone-folded LO phonons (ZFLOs).

Old Text: The negative dispersion of the LO phonon ensures that these weak phonon modes exist within the Reststrahlen band, co-existing in frequency with propagating or localised SPhPs.

New Text: The negative dispersion of the LO phonon ensures that these weak phonon modes exist within the Reststrahlen band. This allows them to coexist in frequency with the propagating or localised SPhPs extant at the Γ point

3. It would be really helpful if the authors could add to Fig. 1b the dispersion of both TO phonons and that of phonon-polaritons (for a single SiC-air interface). Otherwise, it is difficult to have a feeling regarding the hybridization of the modes (“strong coupling”) that the authors are aiming to study.

We included the TO phonon dispersion in the revised manuscript as requested. Concerning the single-interface SPhP instead we could not directly add its dispersion to the figure because it disperses strongly over small, photonic wavelengths near to the Gamma point. As such the most important details of the dispersion would thus not be visible over the crystal wavelength range shown. We thus decided to instead highlight on the figure the total envelope of the SPhP range around Gamma point.

4. I do not understand the significance of Eqs. (1)-(3) for the paper. How do the authors make use of them? What does the displacement X (and equation for it) help me to learn/understand? Initially, I thought that Eq. (3) was used to extract the nonlocal dielectric function $\epsilon_{ps}(\omega, \mathbf{k})$. In this case, maybe it would prefer to see the explicit expression for $\epsilon_{ps}(\omega, \mathbf{k})$, rather than the equation for X.

Otherwise, for instance, I do not see any connection between Eq. (4) and Eq. (3).

We apologise if the previous version of the manuscript lacked in clarity. A lot of the explanation was hidden in the Supplementary Information and could certainly be better referenced in the main body of the manuscript. The purpose of equations 1-3 (1-2 and S2 in the novel version) is as follows. Equation 1 shows the effect of including dispersion on the ionic equation of motion, through the terms proportional to the velocities. Equation 2 (S2 in the novel version) just defines the polarisability utilised in equation 1 in terms of basic quantities which the reader can be expected to understand.

Equation 3 (2 in the novel version) is derived in the Supplementary Information. It illustrates that the solution to equation 1, that is the equation of motion of the ions including spatial dispersion, is a mixed excitation comprised of longitudinal and transverse parts. This is included in the paper as the concept is key to the later results, where we explicitly observe this mixing.

Deriving the spatially dispersive dielectric functions as the Referee suggests can be done directly from Eq. 1 by following the segregation procedure outlined in the Supplementary Information however this alone does not lead to the result in equation 4 (3 in the novel version). This is because, in addition to a change in the dielectric function, the mixing also depends on the wavevector components parallel to and perpendicular to the interface for both the transverse and longitudinal mode through the application of the electrical and mechanical boundary conditions. This means we cannot relate Eq. 4 (3 in the novel version) to the standard Fresnel relation by simply substituting the spatially dispersive dielectric function. We added the spatially dispersive dielectric function to the main body of the manuscript to aid the reader in understanding the physical consequence of the spatially dispersive terms in Eq. 1.

To address the other concerns of the Referee, in the manuscript we also removed equation 2, defining this instead in supplementary equation S2. We made the following change to the text around what was Eq. 3 (2 in the novel version) in order to allow the reader to better understand our motivation:

Old Text: In the Supplemental Information this equation of motion, in conjunction with Maxwell equations and the appropriate electromagnetic and mechanical boundary conditions, are solved by the introduction of auxiliary scalar and vector potentials $\mathbf{Y} = \nabla \cdot \mathbf{X}$, $\mathbf{\Gamma} = \nabla \times \mathbf{X}$, allowing us to write the ionic displacement as

New Text: In the Supplementary Information this equation of motion, in conjunction with the Maxwell equations, is solved by the introduction of auxiliary scalar and vector potentials $\mathbf{Y} = \nabla \cdot \mathbf{X}$, $\mathbf{\Gamma} = \nabla \times \mathbf{X}$. This allows us to write the total ionic displacement as a hybrid, containing both transverse and longitudinal components whose mixing will be instigated by application of the appropriate mechanical and Maxwell boundary conditions

Additionally we explain in the section on SPhPs why the result cannot be replicated by simply using a dispersive dielectric function. We added the following sentences to the SPhP section:

New Text: It is important to note that Eq. 3 cannot be reduced to a simple equation where the dispersionless dielectric function $\epsilon(\omega)$ is replaced by the spatially dispersive $\epsilon(\omega, \mathbf{k})$ derivable from Eq. 1. This is because additionally to the effect of spatial dispersion calculating the reflectance now involves application of both mechanical and Maxwell boundary conditions, resulting in a mixing dependent on the wavevectors on each side of the boundary.

5. I was confused with the notations for the wavevectors. There are 4 of them: \mathbf{k} , k_x , k_B and k_T . I guess with k_B and k_T the z-component is meant. But then I would define them as k_{zB} and k_{zT} , explicitly writing up the expressions for them. Does \mathbf{k} present the x-component of the wavevector? Then I would simply use k_x . Actually, the notation k_x is used in Eq. (5), so are \mathbf{k} and k_x equivalent?

The quantity \mathbf{k} referred to the vectorial wavevector of the transverse mode. We renamed this quantity k_T and defined it in the manuscript. We agree regarding the definitions of k_T and k_B . To this end we replaced them as requested with k_{zT} and k_{zB} . To avoid any confusion we also added two new equations (Eq. 4 and 5 in the novel version) defining the wavevectors in terms of other known quantities.

6. In which equation do the authors make the shift of k_x , specified by the Eq. (5)? Do they substitute Eq. (5) into Eq. (4)? This is not mentioned and should be clarified.

As explained in the Supplementary Information, the shift is only carried out for the longitudinal component of the mode and is therefore essentially encoded inside the function Ω . This is clarified in the body as follows:

Old Text: We can apply this result to the description of the ZFLO modes observed in planar reflectance measurements of a-cut SiC polytypes by shifting the longitudinal in-plane wavevector to...

New Text: We can apply this result to the description of the ZFLO modes observed in planar reflectance measurements of a-cut SiC polytypes by shifting the longitudinal in-plane wavevector to ...while leaving the wavevectors of the transverse components unchanged.

We additionally changed the notation in Eq. 6 (7 in the novel version) to make it clear that this referred only to the longitudinal in-plane wavevector.

7. I do not understand what is meant by the phrase “the anisotropy is reintroduced by considering the transverse wavevector to be that of the extraordinary wave in the crystal”. First of all, is Eq. (4) written for the isotropic half-space? This is not mentioned. Second, I do not understand what the “reintroduction” of the anisotropy means and to which equation/equations it should be applied. Why doing this reintroduction? More details are needed in my opinion.

We agree with the Referee that this point was unclear. We added a detailed explanation of this point at the end of section II of the Supplementary Information, which we hope the Referee will now find clearer. We also made the following change in the main body of the manuscript:

Old Text: The result for a 4H-SiC substrate is shown in Fig 2a, where the anisotropy is reintroduced by considering the transverse wavevector to be that of the extraordinary wave in the crystal...

New Text: The result for a 4H-SiC substrate is shown in Fig 2a. The anisotropy of such a polytype can be taken into account by noting that the leading terms in the numerator and denominator of Eq. 4 when neglecting the ZFLO mode ($\Omega = 0$) are just those from the Fresnel coefficient of an isotropic halfspace. We can then replace the term for the lower halfspace with that derived for an uniaxial halfspace in the local approximation by considering the transverse wavevector to be that of the extraordinary wave in the crystal

8. As far as I understand, the authors fail in developing the theory/simulations for their pillar structure because of the complexity of the problem. But is there any chance for the authors to complement their experimental results with some simplified modelling? For example, at least showing the monopole mode of a single rod (in the local approximation). Then the bandstructure of the pillar lattice can be roughly represented by the back-banding of the dispersion of the monopole mode. Otherwise, it is really difficult to interpret the results presented in Fig. 3 and be convinced that the author's arguments are valid. With many of commercially-available (and also free) simulators, based on the finite-elements methods, it does not seem to be very demanding to simulate an infinite periodic array of rods in the local approximation. Actually, the nonlocal approximation can be also roughly taken into account, by plugging in into Eq. 14 (from the supplement) the momentum of the hybridized mode, estimated from the experiment.

As we noted in our reply to point 2 the pillar lattice is not that which induces the folding and its dispersion is not helpful in describing the back-bending. The Referee is correct to point out that the pillar on substrate system is easy to simulate using finite element methods. This has been studied in a few papers in the past using a combination of experimental and numerical methods by ourselves (Chen et al. ACS Photonics **1**, 718 (2014), Gubbin et al. Physical Review Letters **116**, 246402 (2016), Gubbin et al. Physical Review B **95**, 035313 (2017)) so the red-shift with increasing pitch is well understood. We added a sentence pointing this out in the paper.

The Referee's second suggestion, of using the nonlocal approximation by taking the nonlocal dielectric function and plugging it in the folded wavevector, in our opinion leads to incorrect results. Although an approximation of k to that at the Brillouin zone edge in the dielectric function is reasonable, just making this approximation will not capture the physics we are interested in.

The mixing depends not just on the phonon frequencies at the Brillouin zone edge but also on the coupling of the phonons here with those at the zone centre. This coupling is mediated through the mechanical boundary conditions, depending on the wavevector components and spatial distribution of the transverse fields. The transverse mode of the micropillar contains many wavevectors and is highly inhomogeneous. This means that the only way to arrive at a solution is through coupling Maxwell's equations to those for the displacements subject to the boundary conditions and solving everything together. This is a computationally difficult problem as the hybridisation dramatically increases the number of degrees of freedom in the numerical problem. In order to better explain this point we added this paragraph in the manuscript:

New Text: In order to tackle the problem numerically the electromagnetic fields, described by Maxwell's equations, must be coupled through the mechanical boundary conditions to the ionic equation of motion in Eq. 1. In contrast to the planar case considered in the previous section where solutions were Bloch modes for which the in-plane wavevector was a good quantum number, the micropillar modes contain many wavevectors whose values will be affected by their coupling to the longitudinal degrees of freedom as in plasmonic nonlocality. Given the importance of mechanical boundary conditions for the longitudinal-transverse hybridization, prohibitively expensive full 3D simulations of the coupled electromagnetic and material displacement fields are thus necessary to describe LTPP in the resonator array on a substrate system.

9. The authors mention first the “monopole mode”, but then talk about interpillar dipole-dipole coupling. It sounds a bit contradicting. Shouldn't it be the monopole-monopole coupling?

We agree this could be confusing for people who are not very familiar with the literature on SPhPs. This is the terminology usually used in studies of these SPhP resonators. To help a more general audience we removed the reference to dipole-dipole coupling, replacing it with interpillar coupling.

10. What are the vertical dashed line in Figs. 3a,b? Do they indicate the position of the monopole resonance. If so, how are the authors can be sure about that without having any reference (for example, from the simulations).

We state in the text that the horizontal dashed line is the position of the weak phonon resonance, a known quantity not derived from simulations. The vertical dashed line is merely a guide to the eye, representing the point where the magnitude of the reflectance dips cross in the lower panel.

[1] Ulrich, C., Debernardi A., Anastassakis, E. Syassen, K., & Cardona, M. Raman Linewidths of Phonons in Si, Ge, and SiC under Pressure. *phys. stat. sol. (b)*, 211: 293-300 (1999).

Reviewers' Comments:

Reviewer #2:

Remarks to the Author:

The authors have amended the text and their paper should now be published in Nature Communication.

Reviewer #4:

Remarks to the Author:

I have gone through the paper, the comments by the referees and the responses.

Overall this manuscript is very interesting and presents a timely idea. However, the abstract claims considerably more than what has achieved in the paper.

1) There is no demonstration of electrical pumping or far-field emission. In fact, there is no theory to describe this process either.

2) The only experimental plot is in figure 3. Its unclear how it connects to theory even after the new changes.

3) The authors should add a schematic of what is hybridizing to explain the central result to the broad audience of Nature Communications

4) For their spatially dispersive dielectric constant, what additional boundary conditions do they use?

The paper can be published once the abstract and introduction are reflective of what is directly achieved in the paper.

Answer to the Reviewer #4

We thank this Reviewer for the time and effort they put into reviewing our work. We are pleased that they found our paper worth of publication and in the following we answer point-by-point to the few comments they raised.

Overall this manuscript is very interesting and presents a timely idea.

We thank the Reviewer for their praising words.

However, the abstract claims considerably more than what has achieved in the paper.

We thank the Reviewer for this criticism. Our intention was to present our results in the context of possible future applications, in order to better communicate their relevance and potential impact. Probably this led to an abstract in which the distinction between the results and the future applications was somewhat blurred. We think this problem has been solved in the novel version of the manuscript.

1) There is no demonstration of electrical pumping or far-field emission. In fact, there is no theory to describe this process either.

We agree with the Reviewer that those elements are not present in our manuscript.

We modified the abstract and the introduction trying to make a clear distinction between what we achieved, and what we didn't.

Still we would like to point out that we do perform far-field measurements which implies that phonon-polaritons, if excited, would emit in the far field.

2) The only experimental plot is in figure 3. It's unclear how it connects to theory even after the new changes.

We added a paragraph in the novel version of the manuscript, explaining in clearer terms the link between theory and experiments. Such a link is now spelled out in very explicit terms: our theory predicts an anticrossing for the planar system, and the micropillar system is described by the

same physics. We thus expect to also observe a spectral anticrossing, only above the light-line and thus visible in the far-field.

3) The authors should add a schematic of what is hybridizing to explain the central result to the broad audience of Nature Communications

The typical schematic used to explain hybridization and the formation of polaritons out of the bare modes is already present in the manuscript (Fig. 2b), although the previous versions of the manuscript only spent a few words on it. In the novel version we added a paragraph which describes the essence of the hybridization process by referring to such a figure.

4) For their spatially dispersive dielectric constant, what additional boundary conditions do they use?

We use the boundary conditions describing the continuity of the mechanical forces, detailed in Eq. 33 of the Supplementary Material.